# Soft elasticity optimises dissipation in 3D-printed liquid crystal elastomers

D. Mistry [1,4✉], N. A. Traugutt [1,5], B. Sanborn [2], R. H. Volpe[3], L. S. Chatham[3], R. Zhou[1], B. Song[2], K. Yu[1✉], K. N. Long [2] & C. M. Yakacki [1,3✉]

Soft-elasticity in monodomain liquid crystal elastomers (LCEs) is promising for impact-absorbing applications where strain energy is ideally absorbed at constant stress. Conventionally, compressive and impact studies on LCEs have not been performed given the notorious difficulty synthesizing sufficiently large monodomain devices. Here, we use direct-ink writing 3D printing to fabricate bulk (>cm$^3$) monodomain LCE devices and study their compressive soft-elasticity over 8 decades of strain rate. At quasi-static rates, the monodomain soft-elastic LCE dissipated 45% of strain energy while comparator materials dissipated less than 20%. At strain rates up to 3000 s$^{-1}$, our soft-elastic monodomain LCE consistently performed closest to an ideal-impact absorber. Drop testing reveals soft-elasticity as a likely mechanism for effectively reducing the severity of impacts – with soft elastic LCEs offering a Gadd Severity Index 40% lower than a comparable isotropic elastomer. Lastly, we demonstrate tailoring deformation and buckling behavior in monodomain LCEs via the printed director orientation.

[1] Department of Mechanical Engineering, University of Colorado Denver, Denver, CO 80204, USA. [2] Materials and Failure Modeling Department, Sandia National Laboratories, Albuquerque, NM 87123, USA. [3] Impressio Inc., 12635 E. Montview Blvd, Suite 214, Aurora, CO 80045, USA. [4] Present address: School of Physics and Astronomy, University of Leeds, Leeds LS2 9JT, UK. [5] Present address: Desktop Health, 63 3rd Avenue, Burlington, Massachusetts 01803, USA. ✉email: d.a.mistry@leeds.ac.uk; kai.2.yu@ucdenver.edu; chris.yakacki@ucdenver.edu

One of the most exciting but often overlooked applications for liquid crystal elastomers (LCEs) is for use in strain-rate-dependent impact absorbing devices[1–5]. In 2001, Clarke et al. reported that LCEs—which incorporate the anisotropic ordering of liquid crystals into elastic polymer networks—demonstrate elevated loss tangents ($\tan(\delta)= G''/G'$) as high as 1.5 when held at temperatures between their glass transition and nematic-to-isotropic transition temperatures ($T_g$ and $T_{NI}$, respectively)[6]. These values of $\tan(\delta)$ correspond to highly viscous and dissipative materials and are far greater than values (~0.1)/ found in traditional isotropic elastomers[7,8].

Despite this exceptional mechanical behavior, there have only been a handful of published papers concerning LCE dissipative mechanical properties[1,2,4,6,9–11]. Azoug et al. reported on the large and strain-rate-dependent tensile hysteresis between loading and unloading stress-strain curves for polydomain (macroscopically unaligned liquid crystallinity, Fig. 1a) LCEs[9]. This hysteretic behavior describes an efficient dissipator of strain energy in a material that can either be plastic or elastic depending on the network structure[5]. Moreover, using digital light processing 3D printing, our group recently showed that polydomain LCEs had superior energy dissipation and rate dependency behavior compared to conventional isotropic elastomeric materials in compression[1]. Curiously, this work showed that the well-known soft-elasticity of polydomain LCEs under tension is not seen in compression—a result also recently reported by Shaha et al[12].

Soft elasticity refers to the unique plateau-like tensile mechanical response of LCEs as described by theory pioneered by Warner and Terentjev (Fig. 1b). This load curve shape bears a resemblance to that of an ideal dissipator[13], i.e., a long plateau of constant and finite stress, and is therefore a phenomenon one would hope to observe and exploit in compression to create highly hysteretic and ideal dissipators of impact energy[14]. Interestingly, Shaha et al. observed soft-elastic behavior in compression for uniaxial monodomain (macroscopically aligned and anisotropic liquid crystallinity, Fig. 1a) LCEs compressed parallel to the molecular symmetry axis, known as "the director." Thus, monodomain LCEs have the potential to act as efficient dissipators of impact energy.

Historically, fabricating bulk monodomain LCE devices large enough for impact-absorbing applications has been challenging[15,16]. For instance, the two-step LCE synthesis process used by Shaha et al. has limited scalability in producing bulk (>cm³) and uniform devices. However, the recent application of direct-ink writing (DIW) 3D printing to the fabrication of LCE devices offers a new route to producing arbitrarily sized monodomain LCEs. In DIW, an LC oligomer is shear aligned into a monodomain when extruded through a nozzle—a state which is then fixed by subsequent photo-crosslinking of the oligomers into a network (Fig. 1c)[15]. Line-by-line and layer by layer, macroscopic monodomain LCE devices can be fabricated with anisotropy controlled via the chosen movements of the print head (Fig. 1d). Despite the obvious potential of DIW printing to fabricate bulk monodomain LCE devices, existing studies have only been able to produce thin devices, <10 printed layers or <2 mm thick—studied for their shape actuation properties[17–23].

In this paper, we DIW print bulk monodomain LCE devices up to $12 \times 12 \times 7.5$ mm³ in size and investigate their anisotropic and dynamic mechanical responses in comparison to equivalent polydomain LCEs and conventional isotropic materials. Through

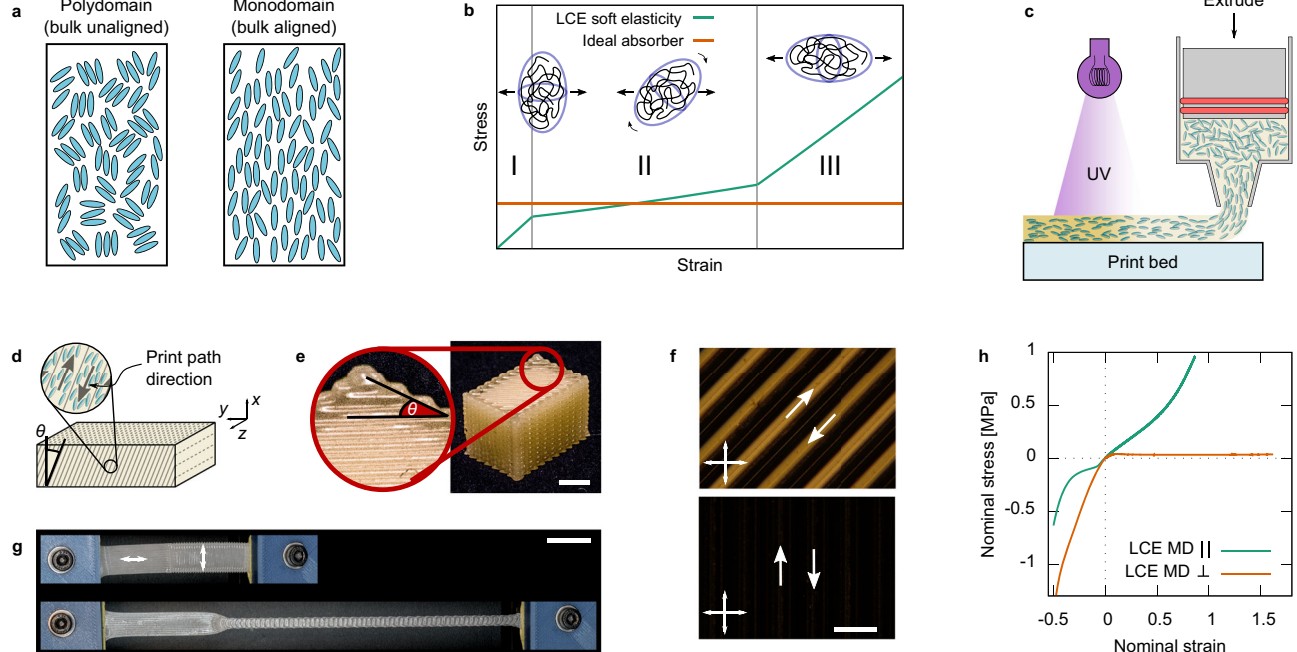

**Fig. 1 DIW 3D printing setup and basic material properties. a** Illustrations of the LC-molecular alignment in polydomain and monodomain devices. **b** The rotation of the anisotropic polymer conformation gives rise to the tensile soft-elastic response of LCEs, which bears resemblance to the idealized load curve of a strain-energy absorbing device. **c** Liquid crystal oligomers are shear aligned when extruded through the 3D printer's nozzle and photocrosslinked into an elastomer. **d** The direction of print head movement dictates the orientation of the liquid crystal director, thus arbitrarily aligned devices can be constructed. **e** An example of a bulk 3D-printed LCE, optimization of the print conditions allows high-quality printing of bulk (>cm³) devices. Bar = 5 mm. **f** Crossed-polarizing microscopy of a single printed layer. The uniformity of each image and high contrast between them is indicative of excellent liquid crystalline alignment within printed lines. In the top figure, the striped appearance is caused by cylindrical profile of printed light lensing the transmitted light. Bar = 0.5 mm. **g** Mechanical anisotropy and soft-elastic response of the LCE visually demonstrated by straining a bi-strip with domains of parallel and perpendicular orientation. Bar = 10 mm. **h** The tensile and compressive mechanical anisotropy of printed LCEs (tested separately). The soft-elastic response is seen when perpendicular (parallel) oriented samples are stretched (compressed).

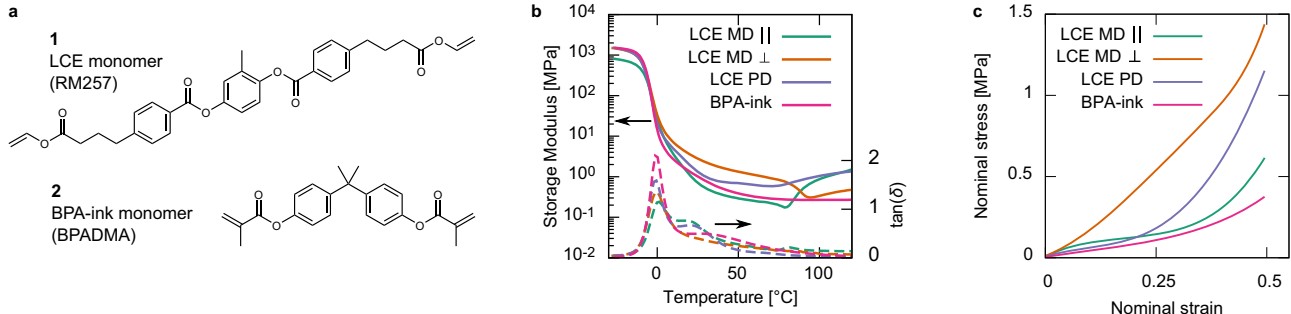

**Fig. 2 Key chemical structures and basic mechanical characterization. a** Chemical structures of the diacrylate monomer groups used in this work. RM257, **1**, has a stiff core which promotes liquid crystalline ordering. Bisphenol-A dimethacrylate (BPADMA, **2**) does not have a rod-like core structure. **b** Dynamic mechanical analysis data comparing the storage moduli (solid) and loss ratio, tan($\delta$), (dashed) properties of the key materials compared here. **c** Quasi-static ($10^{-4}\,s^{-1}$ displacement rate) compressive load curves of all materials compared here. Despite the key structural differences between the LCE MD ∥, LCE PD LCE BP-ink elastomer, they all have similar low-strain behavior meaning they can be reasonably compared.

quasi-static, high strain rate, and impact testing experiments, we show that soft-elasticity in monodomain LCEs offers a fundamentally unique route to enhancing the impact-absorbing behavior of solid elastomers and which is simply above and beyond that which can be achieved with conventional isotropic elastomers. Additionally, we show that by programming the print direction within LCE devices, we can introduce and control the nature of buckling responses during compression—thus opening additional routes to controlling and optimizing the dissipation of mechanical energy.

## Results

**DIW printing and materials overview**. Large monodomain LCE blocks were DIW-printed after optimizing the printing parameters (Fig. 1e). By using custom G-code scripts to print series of parameter-tuning matrices (detailed in methods) we selected the proper ink temperature, extrusion rate, print speed, layer height, and extrusion width needed to stably print devices of over >20 printed layers[15]. Gel fraction tests on DIW-printed LCEs gave insoluble fractions of 99.3 ± 0.4%—confirming the printed materials were near-completely crosslinked. Fabricating devices on this scale (i.e. devices >5 mm thick) is key to our mechanical and impact studies as well as the end-application of LCEs as impact-absorbing materials. The monodomain liquid crystalline alignment of the printed LCE devices is readily seen when viewing a single printed layer (310 μm thick) via crossed-polarizing microscopy. When the print direction is oriented at 45°, a highly uniform bright state is seen (Fig. 1f, top). In the figure, the adjacent printed lines are in contact with one another. The black bands seen are a consequence of the printed lines' cylindrical profile (a result of the circular nozzle outlet) lensing the transmitted light[15]. When the print direction is oriented parallel with one of the polarizers, a dark state is seen (Fig. 1f, bottom). The appearance of the material and contrast shown in these images indicate a highly uniform state of uniaxial alignment in the printed LCE.

The local mechanical response of DIW-printed LCE devices can be tuned through the design of the printing direction. A bi-strip device (Fig. 1g) demonstrates the ability to tune the anisotropic mechanical response of LCEs using DIW printing. Upon straining the device, the perpendicularly oriented region is deformed to a far greater extent than the parallel oriented region —a consequence of soft elasticity.

The magnitude of the DIW-printed LCE's mechanical anisotropy is quantified by the load curve of Fig. 1h. The figure plots together result from tensile mechanical tests and separate compressive

mechanical tests (see methods). Tensile strains (upper right quadrant) applied parallel to the print direction (and director) give rise to a conventional elastomeric response. In comparison, for tensile strains up to 1.5 (experimental limit) applied perpendicular to the director, the LCE demonstrates a near-zero modulus across the soft elastic plateau. In compression (lower left quadrant), a similar anisotropic response is seen but with two differences. First, the soft-elastic (classical elastic) response is now seen for strains parallel (perpendicular) to the director. Second, the apparent magnitude of anisotropy is less than that seen in tension—as the maximum possible nominal strain in compression, −1, is far less than the strains experienced in tension. The significant levels of anisotropy and the soft-elastic effect in the bulk DIW-printed LCE devices again indicate high levels of liquid crystal alignment throughout each layer of our printed device.

Here, we compare the dissipative and impact-absorbing capability of our monodomain (MD) LCE in 3D-printed devices when compressed parallel (LCE MD ∥) and perpendicular (LCE MD ⊥) to the director and print direction. We also compare the response of a molded polydomain LCE (LCE PD) of similar chemistry and a DIW-printed isotropic elastomer (BPA elastomer) for further comparison and discussion. The liquid crystallinity of our LCEs results from the use of the diacrylate monomer, RM257 **1** (Fig. 2a), which is has a central stiff and rod-like core. Our isotropic elastomer material replaces RM257 with bisphenol-A dimethacrylate (BPADMA, **2**, Fig. 2a), which features similar chemical groups as RM257 but does not have a rod-like core.

The LCE PDs and BPA elastomer's crosslink densities were tailored to ensure comparable thermomechanical properties as the DIW printed LCEs. Figure 2b shows the storage moduli and tan($\delta$) of each material measured using dynamic mechanical analysis (DMA). The peak of each material's tan($\delta$) shows that all materials have glass transition temperatures ($T_g$) within 1 °C of each other. Additionally, in the rubbery regime above $T_g$, these materials have comparable storage moduli. At room temperature (20 °C), the monodomain LCE has an anisotropic storage modulus of 1.1 and 3.6 MPa for small strains parallel and perpendicular to the director, respectively. The storage moduli of LCE PD and the BPA elastomer lies between the values for the monodomain LCE at 2.0 and 1.4 MPa, respectively. The tan($\delta$) curves also show that these materials have elevated tan($\delta$) plateaus above $T_g$. While this is expected for LCEs[10], an elevated tan($\delta$) was not expected for the BPA elastomer. Overall, the high tan($\delta$) of the BPA elastomer makes this an excellent material to compare to the LCEs, allowing us to directly compare the influences of viscoelasticity alone (BPA) versus viscoelasticity coupled with mesogen rotation (LCE).

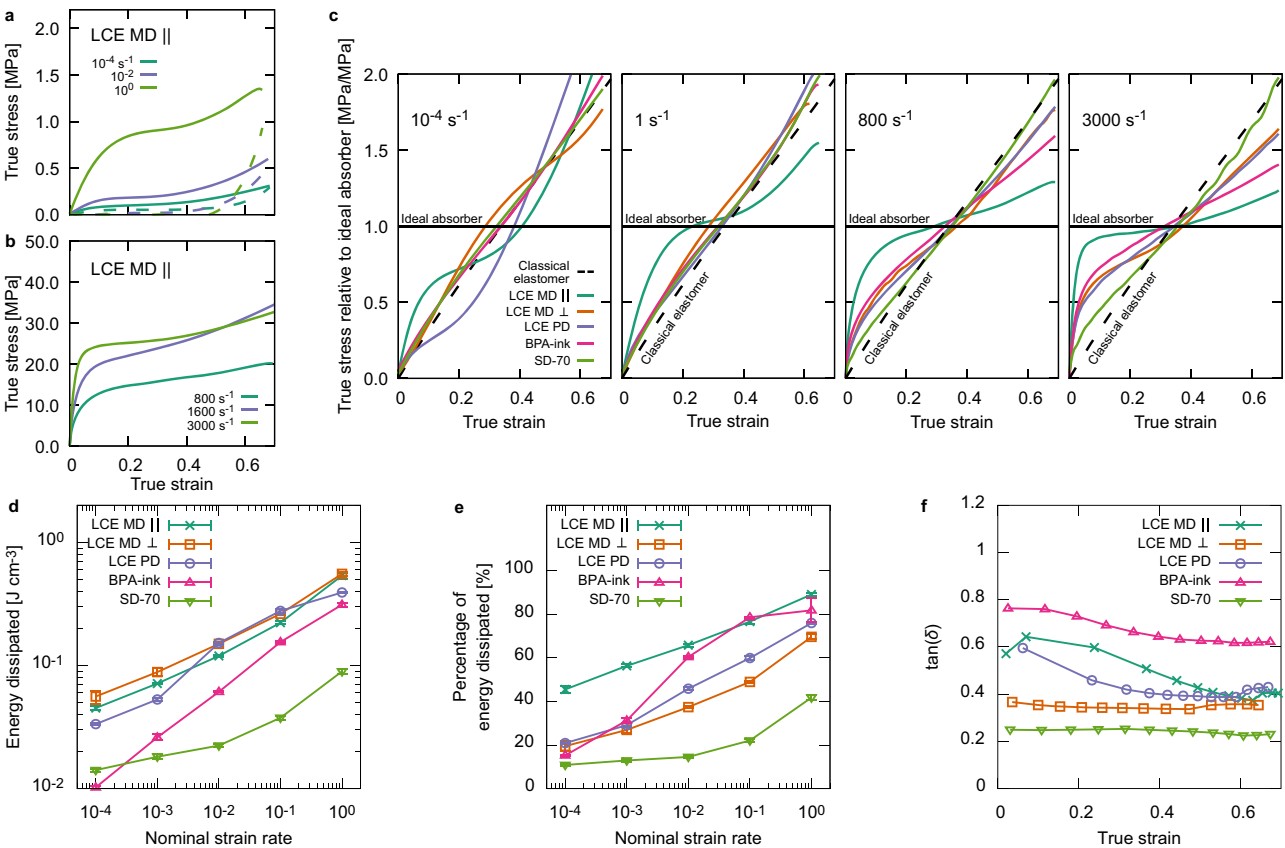

**Fig. 3 Strain energy absorption. a** Examples of compressive loading (solid) and unloading (dashed) curves for the LCE MD ‖ with nominal strain rates between the quasi-static $10^{-4}$ s$^{-1}$ and the intermediate $10^{0}$ s$^{-1}$ demonstrating the rate-dependence of the soft-elastic response. The area between the load curves corresponds to dissipated strain energy. **b** Examples of impact-rate compressive load curves, measured using a Kolsky (Split Hopkinson) bar, the rate dependency is still evident with the LCE demonstrating stiffer (softer) elastic behaviors at low (intermediate) strains for increasing displacement rate. **c** Comparisons of each material's compressive behavior at various strain rates relative to the constant-stress behavior of the ideal dissipator of strain energy. For reference, the expected response of a classical entropic elastomer is also shown. Representative curves are shown from a testing size of at least three experiments. **d** Quantified measurements of the dissipated energy and (**e**) percentage dissipated energy relative to loaded energy for each tested material from quasistatic $10^{-4}$ s$^{-1}$, and the intermediate $10^{0}$ s$^{-1}$ nominal strain rates. **f** The loss ratio, tan($\delta$)=$E''/E'$, from small strain (0.1% amplitude) dynamic mechanical analysis tests performed on each sample at different levels of compressive strains. For (**d**) and (**e**) errors represent SD, for (**f**), the SD experimental errors are smaller than the points.

In addition to the materials already described, we also compare these materials against Sorbothane® Durometer-70 (SD-70, durometer measured on the Shore 00 scale), a commercial polyurethane-based impact absorbing and vibration-isolating material. The compressive load curves in Fig. 2c show that all materials all have comparable elastomeric behavior at the quasi-static nominal strain rate ($10^{-4}$ s$^{-1}$).

**Rate dependency of load curve shape and energy dissipated**. Compressive load curves for LCE MD ‖ over 8 decades of nominal strain rate tested show the material's highly rate-dependent and soft-elastic behavior (Fig. 3a, b). For rates between $10^{-4}$ and $1$ s$^{-1}$ we used a uniaxial testing machine, for rates above $800$ s$^{-1}$, we used a Kolsky (split-Hopkinson) bar. In each case, samples were loaded to compressive true strains of $-0.7$ (nominal strains of $-0.5$, note compressive strains are shown as positive for simplicity). Samples used had a low aspect ratio to avoid any buckling of the samples during testing (see "Methods"). True stresses were calculated using a constant volume and ideal deformation (minimal edge effects) assumption by multiplying the nominal stress by the deformation of $\lambda = \epsilon_N + 1$, where $\epsilon_N$ is the nominal strain. We note that these assumptions have their limitations for the presented DIW printed LCEs and BPA

elastomers, which have porosities of 18% and 9%, respectively (Supplementary Figure 1). These devices' porosity is caused by the cylindrical profile/of the extruded inks (a consequence of the circular nozzle outlet) trapping parallel channels of air inside the printed devices. Despite this, these calculations of true stress still provide a realistic insight into the tested materials' mechanical responses, which undergo significant increases in cross-sectional area upon compression. We note that the porous channels do not introduce any anisotropy in the character of the BPA material's dynamic and quasi-static mechanical behavior (Supplementary Fig. 2b).

Figure 3c shows loading curves for each material at nominal strain rates of $10^{-4}$, $1$ ($10^{0}$), $800$ ($8 \times 10^{2}$), and $3000$ ($3 \times 10^{3}$) s$^{-1}$. In these figures, the true stresses have been normalized against the stress level of an ideal absorber of strain energy (see "Methods"). Presenting the data in this way enables comparison of each material's stress-strain response at each nominal strain rate and how the load curves' characteristic behavior changes with increasing nominal strain rate. For reference, on each graph we also show the expected behavior of an ideal absorber and for a classically elastic and volume conserving material (an incompressible neo-Hookean solid), described by

$$\sigma_T = \mu\left(e^{2\epsilon_T} - e^{-\epsilon_T}\right), \qquad (1)$$

where $\mu$ is the usual characteristic rubber modulus, and $\sigma_T$ and $\epsilon_T$ are the true stress and strain, respectively.

From the quasi-static rate of $10^{-4}\,s^{-1}$ we can identify three types of material behavior. The first type is classical elasticity from materials that did not undergo mesogen rotation (LCE MD $\perp$, BPA elastomer, and SD-70). The second is polydomain soft-elasticity displayed by LCE PD. Some soft-elastic effects are visible as the material softens from the classical behavior at a strain of ~0.1, which contrasts our previous work and that of Shaha et al. Despite demonstrating a small amount of soft elasticity, the LCE PD displays the worst performance of all materials tested when compared to the ideal absorber. The last type of material behavior is monodomain soft-elasticity with optimized mesogen rotation, shown by LCE MD $\parallel$. In this behavior, the material is initially stiffer than a conventional elastomer and then softens in a plateau – resulting in the closest performance to an ideal absorber.

By the intermediate strain rate of $1\,s^{-1}$, the LCE PD sample no longer displays any soft-elasticity. Therefore, LCE PD behaves, like LCE MD $\perp$, BPA, and SD-70, as a classical elastomer—not-optimized for absorbing mechanical energy. By contrast, the rate dependence of the LCE MD $\parallel$ further accentuates the soft-elastic effect and brings the material's performance closer to that of an ideal absorber.

At impact rates of 800 and $3000\,s^{-1}$, LCE MD $\perp$, LCE PD and BPA-ink all have similar, slightly improved, responses departing from that of a classical elastomer. However, the LCE MD $\parallel$ still shows a fundamentally different response that is again much closer to ideal behavior. These trends continue to the fastest nominal strain rate tested ($3000\,s^{-1}$), where the LCE MD $\parallel$ is converging toward the ideal response, a remarkable behavior for a non-foam material. For additional comparison, we also highlight Wang et al.'s similar experiments performed on a polyurea, a material type commonly used in blast protection applications[24]. Their results at impact rates ($3300\,s^{-1}$ and greater) show that at low true strains (<0.2), polyureas demonstrate improvements similar to LCE MD $\parallel$ in their load curve shape with increasing nominal strain rate. However, at true strains greater than ~0.2, the polyurea showed significant stiffening and appeared to "bottom out" at relatively low strains. Therefore, monodomain soft-elastic LCEs show enhanced characteristics for energy absorption high-rate impacts compared to common incumbent materials.

In addition to the loading and storage of energy during a compressive deformation, for impact-absorbing applications it is also important to consider the magnitude (Fig. 3d) and percentage (Fig. 3e) of loaded energy that is dissipated. For example, a highly hysteretic and dissipative material is not necessarily a more optimal impact-absorber when compared to an elastic material if it is significantly softer and therefore does not dissipate a large enough magnitude of energy for a given application.

Figure 3d shows that from rates of $10^{-4}$ to $1\,s^{-1}$, the LCE samples dissipate similar magnitudes of strain energy and have similar rate dependencies. Even though the LCE MD $\parallel$ has the softest compressive response (Fig. 2c), it dissipates a comparable magnitude of energy as the stiffer LCE samples due to its highly soft elastic and hysteretic behavior (Fig. 3a). Furthermore, the LCE MD $\parallel$ demonstrates this similar performance while minimizing peak stresses and acting the closest to an ideal absorber across strain rates (Fig. 3c). In comparison to the LCEs, the BPA elastomer dissipates a significantly lower magnitude of energy at the quasistatic rate of $10^{-4}\,s^{-1}$ (~25% of the LCE's dissipatation); however, this increases with a greater strain rate dependency (~50% of that LCE MD $\parallel$ and LCE MD $\perp$ dissipate at

$1\,s^{-1}$). At $10^{-4}\,s^{-1}$, SD-70 also dissipates significantly less energy than the LCEs, but unlike the BPA-elastomer this disparity increases with strain rate.

Figure 3e demonstrates how LCE MD $\parallel$ dissipates the highest percentage of strain energy. Across strain rates, and particularly at low rates, LCE MD $\parallel$ dissipates a far greater proportion of the loaded energy, from $45 \pm 1\%$ at a rate of $10^{-4}\,s^{-1}$, rising to $89.1 \pm 0.8\%$ at $1\,s^{-1}$. By comparison, LCE MD $\perp$ and LCE PD both dissipate ~20% and ~73% of loaded energy at rates of $10^{-4}$ and $1\,s^{-1}$, respectively. While at room temperature, these slower rates do not relate to impact conditions, they provide an insight to the enhanced dissipative behavior at higher temperatures via the time-temperature superposition principle. The BPA shows a curiously different response. Looking back to Fig. 2c, we see that at quasi-static nominal strain rates, the BPA elastomer has a similar stress-strain loading curve as LCE MD $\parallel$ (i.e. it loads a similar magnitude of energy), however, the BPA elastomer instead behaves quite elastically and dissipates only $15.2 \pm 0.3\%$ of the loaded energy. This percentage is similar to LCE MD $\perp$, and LCE PD – which offer little or limited mesogen rotation capability, but is 67% less than LCE MD $\parallel$—which has a large capacity for mesogen rotation (Fig. 3d). However, by $1\,s^{-1}$, the BPA elastomer can dissipate a similar percentage of loaded energy ($82 \pm 6\%$) to LCE MD $\parallel$ ($89.1 \pm 0.8\%$). Figure 3e also shows that SD-70 demonstrates by far the lowest dissipative capacity of all materials tested.

From the load curves, we can deduce that at low strains, LCE MD $\parallel$ can accumulate more strain energy (faster than the material can relax) than any other material tested. The stress level achieved by LCE MD $\parallel$ is then somewhat maintained during the soft-elastic plateau, where there is a greater balance in the rate at which strain energy builds and dissipates. The fast reduction in stress as LCE MD $\parallel$ unloads means LCE MD $\parallel$ had already dissipated much of its stored energy at the point at which unloading begins. While this implies viscous and somewhat liquid behavior, we note that the LCE MD $\parallel$ sample returned to its original shape within a few minutes of each test's completion, and the responses were almost identical upon repeated tests. The ability to dissipate loaded energy quickly and effectively is essential for impact-absorbing applications where any energy not dissipated can be returned as kinetic energy loading to rebounding.

The BPA elastomer's comparatively sharp increase in percentage dissipated energy at a threshold rate of ~$10^{-2}\,s^{-1}$ indicates a change in the thermomechanical response of the BPA elastomer in a way which appears to make it suitable for dissipating mechanical energy. This is attributed to the time-temperature superposition principle, in which an increase in strain rate is shifting the material closer to its glass transition, from an elastomeric to a leathery response with increased viscous effects. However, this also means that at increasing temperatures, one would expect the BPA elastomer's dissipative capability to notably drop off when compared to the LCEs as the material quickly increases in its elasticity.

By comparing the energy dissipated as a function of strain rate (Fig. 3e), we can start to quantify the contributions of mesogen rotation and liquid-crystallinity in elastomers. The LCE materials all have equivalent chemical compositions but different director orientations to the axis of loading. The LCE MD $\parallel$ samples demonstrated the highest percentage of energy dissipated, illustrating the benefit of LCEs aligned for optimized mesogen rotation. When comparing the LCE MD $\parallel$ to non-mesogenic elastomers, the contributions of mesogen rotation can be clearly seen at low-strain rates where viscoelastic effects are minimized. At higher strain rates, the non-mesogenic BPA and SD-70

polymers demonstrate increased viscous effects due to time-temperature superposition and increase energy dissipation. Overall, the tailored structure of the LCE MD ∥ sample provides a superior combination of optimized mesogen rotation and viscoelasticity to demonstrate enhanced energy dissipation across a wide range of test conditions.

This inferred behavior is supported by Fig. 3f, which shows the tan(δ) of each material at various levels of compressive strain. The 1 Hz sinusoidal strains of 0.1% amplitude correspond to a root mean square nominal strain rate of $5 \times 10^{-3}\,\mathrm{s}^{-1}$. The tan(δ) of all the materials tested have differing extents of strain-dependency. For the LCEs, the differences can be linked to the presence of soft-elastic effects. For the non-soft elastic LCE MD ⊥, tan(δ) remains constant with strain at a base level of ~0.35. For the polydomain soft-elastic LCE PD, tan(δ) is initially elevated at ~0.6, but then quickly converges to the base level. The monodomain soft-elastic LCE MD ∥ starts with a tan(δ) similar to that of LCE PD; however, this is maintained for a greater range of strains before converging to the based level. Given that the monodomain soft-elasticity in LCE MD ∥ allows for greater extent of mesogen rotation with strain than the polydomain soft-elasticity in LCE PD, we conclude that mesogen rotation has a significant impact on dissipation in LCEs and enhances the dissipative performance over non-soft elastic (LCE MD ⊥) and conventional (SD-70) elastomers. A notable surprise shown by Fig. 3f is that the BPA elastomer has an exceptionally high tan(δ), between ~0.8 at low strains and ~0.6 at high strains, greater than for any other material tested here. Despite having the highest tan(δ) values throughout compression, the LCE MD ∥ still outperformed the BPA elastomer in terms of magnitude and percentage of energy dissipated, highlighting the importance of optimized mesogen rotation compared to traditional viscoelasticity alone.

**Impact performance**. Rate-dependent compressive mechanical testing has shown that monodomain soft elasticity offers several enhancements in an elastomer's ability to dissipate energy. Next, we consider the performance during impacts simulated by drop testing (Fig. 4). In our tests, samples of approximately $12 \times 12$ mm² cross-sectional area and 7.5 mm height were impacted by 2 kg cylindrical mass of radius 38 mm (see methods). Impact testing is fundamentally different from compressive testing as the strain rate of the sample is not constant throughout the duration of the test. The LCE MD ∥, BPA elastomer, and SD-70 materials are compared to explore how the promising characteristics of monodomain soft elasticity translate to impact behavior. For clarity of discussion and as their structures do not take advantage of soft elasticity under compression, data for LCE MD ⊥ and PD samples are not shown in Fig. 4. However, these data are shown in Supplementary Fig. 4.

For illustrative purposes, Fig. 4a shows the impact response for conventional soft and stiff elastomers compared to ideal behavior. The limitation of conventional elastomers is that they will often slow the impacting object via a sharp peak in acceleration over a narrow strain range. Conversely, the theoretical ideal shock absorber instead provides a constant acceleration over the entire possible strain range, fully compressing to 100%—thus minimizing the peak acceleration experienced. For simplicity, we do not consider the consequences of "jerk"—the rate of change of acceleration with respect to time, except to note that step-like changes in acceleration are not always desirable during impacts.

Figure 4b shows the acceleration experienced by the dropped mass as it impacts and strains the samples. A list of drop heights, impact speeds, energy densities, and initial impact nominal strain rates in these tests is shown in Supplementary Table 1. The LCE MD ∥ demonstrates plateaus in the accelerations at intermediate strains that we can reasonably conclude are manifestations of monodomain soft elasticity in impact conditions. These soft elastic plateaus enable a more uniform deceleration of the dropped mass, giving a response closer to that of the ideal absorber shown in Fig. 4a. Additionally, the acceleration level for the soft-elastic plateau increases with impact energy and acts as a mechanism to offset increases in the peak acceleration. While all materials tested here are clearly viscoelastic (Fig. 3), only LCE MD ∥ translates these rate-dependent effects to their mechanical response in drop tests. This is a unique, passively adaptive mechanism whereby the material behaves stiffer in response to impacts of increasing intensity. By comparison, the BPA elastomer and SD-70 lack any intermediate plateau in accelerations and instead demonstrate a too-soft response as the deceleration is confined to a sharp peak in acceleration at strains of ~0.7 and which undergoes greater increases in peak height compared to the curves for LCE MD ∥.

For head impacts, the severity of an impact and the probability of a person suffering a concussion significantly increases nonlinearly with peak acceleration[25]. These effects can be encapsulated and quantified using the Gadd Severity Index (GSI) which is calculated via the following integral[26]:

$$\int_0^t \mathrm{d}t\, a(t)^{2.5}, \qquad (2)$$

where $a(t)$ is the acceleration of the impact as a function of time. The 2.5 power in the GSI penalizes acceleration peaks and high values. Here, we perform the integral for times from the start of the impact until the velocity of the dropped mass is zero. Each GSI was normalized to the GSI of an idealized absorber (see "Methods").

Figure 4c plots, for each test condition, the GSI against the peak strain. In response to the lowest intensity impacts, the LCE

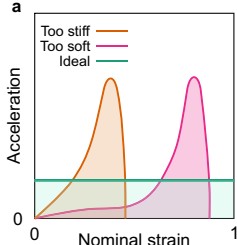
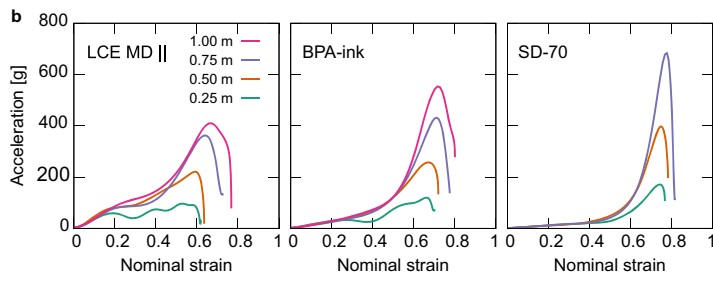
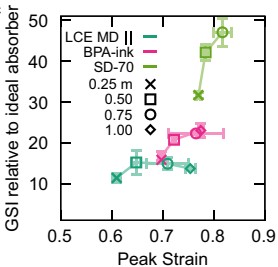

**Fig. 4 Impact performance. a** Illustration of drop test responses for typical non-porous materials of different stiffnesses, and that of the idealized impact absorber which would fully compress at a constant acceleration. **b** Representative readings from an accelerometer mounted on a 2 kg mass as it is dropped from various heights onto devices of LCE MD ∥, BPA elastomer and SD-70. **c** A summary of drop test results in an Ashby-style plot. Plotting, for each material and drop height, the peak strain against the Gadd severity index (Eq. 3) reveals each material's balance of minimizing the severity of impacts and their capability to increase their maximum strain in response to impacts of greater intensity. Errors represent the SD.

MD ‖ undergoes a significantly lower strain (0.6) than the BPA elastomer and SD-70 (0.7 and 0.8, respectively) while also experiencing the lowest GSI—30% lower than that of the BPA elastomer. As the drop height increased, the LCE MD ‖ samples maintained a relatively constant normalized GSI performance, remaining below 16. However, the maximum strain significantly increases by 0.15—the most of any of the materials tested. This increase in maximum strain, along with the increased plateau height, offsets the increase in the peak acceleration and is responsible for the relatively constant GSI performance. At the maximum impact intensity tested, the GSI for the monodomain soft elastic LCE MD ‖ performs 40% better than the BPA elastomer. The lesser performance of the BPA elastomer is quite curious given the DMA data of Fig. 3f which would suggest the material to be an effective absorber of mechanical energy. The resolution of this apparent inconsistency is that while at intermediate-to-high speed deformations the BPA material can effectively dissipate the energy which is loaded, it is fundamentally limited by being too soft early in it deformation and so it fundamentally has limited characteristics suited for safely absorbing impact energy. Lastly, the commercial impact-absorbing material, SD-70, is much too soft to absorb the impact energy of this magnitude as with increasing drop height, the maximum strain barely increases while the peak acceleration, and hence GSI increases significantly—this is characteristic of the material bottoming-out. Note that for SD-70, drop tests from 1.00 m were not performed to protect the experimental apparatus.

**Controlling buckling deformations with anisotropy**. The richness in the mechanical behaviors that can be achieved with anisotropic monodomain LCEs evidently show great promise in impact-mitigating technologies. Until this point, we have solely compared the inherent material characteristics in low aspect ratio devices where buckling is avoided. However, buckling deformations offer additional modes for the dissipation of mechanical energy[27]. We now demonstrate the additional levels of control over compressive mechanical behavior possible in DIW-printed monodomain LCE devices in high aspect ratio devices where buckling effects are likely (see "Methods"). The DIW-printing process imparts liquid-crystal alignment into the material to control anisotropy and buckling behavior (Fig. 5). Three high aspect ratio LCE pillars were tested with the print pathways 0, 45, and 90° to the direction of compression. The photographs show the devices at 0 and 50% nominal strains, and the illustrations trace the director profile via the visible printed lines.

Compressing along the director (Fig. 5a) yields the LCE MD ‖ response seen throughout our results, where the director rotations and possible stripe domains of soft elasticity is analogous to microscopic buckling within the device. Compressing perpendicular to the director (Fig. 5b) initially gives the LCE MD ⊥ response seen in Fig. 3. However, past a nominal strain of 0.25, the high aspect ratio sample undergoes a macroscopic buckling instability, caused by a low shear modulus from the shearing of printed layers along the director. Lastly, for compressions applied at 45° to the director, all buckling effects are suppressed, and the devices demonstrate a more classical elastic response despite its high aspect ratio geometry (Fig. 5c). This additional degree of buckling complexity that one can introduce and control via the print orientations opens the door to mimicking the mechanics of biological materials such as the heterogeneous and anisotropic intervertebral disc[28]. We note that for the illustrations for director orientation in the compressed state for each sample, we are proposing an averaged orientation over a bulk scale and averaged over any stripe domains which may have formed (mostly likely for the compressions parallel to the director).

## Discussion

In this work, we explored the effects of soft elasticity in bulk monodomain LCEs under compression, which has potential use in impact absorbing devices. Through careful optimization of DIW print conditions, we were able to fabricate (to the best of our knowledge) the largest monodomain LCE devices reported to date, and which allowed us to conduct an exceptionally in-depth study of LCE compressive mechanical properties. Our results show that by applying compressions along the director in monodomain LCEs, soft elasticity is observed which translates to enhanced dissipative characteristics over conventional elastomers and polydomain LCEs, like those previously studied by ourselves and others.

Our monodomain soft-elastic LCE devices are capable of dissipating large quantities of strain energy at relatively constant levels of stress – a close-to-ideal behavior which avoids peaks in stress, and almost all of which (90% at a strain rate of $1\,\mathrm{s}^{-1}$) is quickly dissipated. In practically all metrics tested, the LCE MD ‖ samples outperformed a chemically identical LCE loaded perpendicular to the director, a thermodynamically equivalent polydomain LCE, and an isotropic BPA-ink elastomer with a similar glass transition temperature. We attribute the performance of the LCE MD ‖ to its most significant material differences to the other materials – its monodomain and soft elastic nature.

The DMA results for our comparator conventional BPA elastomer succinctly demonstrates how the performance of a materials as dissipators of large-strain or impact energy is not just determined by a the material exhibiting a high tan(δ) throughout its load curve. While this clearly can be highly optimized (the BPA elastomer had the greatest tan(δ) of all materials tested), steps must also be taken to ensure the material loads energy in an optimized manner. Monodomain soft elasticity is evidently an effective mechanism to realize improved impact-absorbing behavior in solid elastomers with the additional rate-dependency offering a material that performs consistently over impacts of differing intensities.

There is much to be explored in future studies. First, we note the LCEs studied here are synthesized from liquid crystalline monomers that were optimized for the displays industry and chosen here (along with the other components) for their low-cost and availability. Future studies should seek to optimize material design and understand how LCE dynamics, i.e. the different relaxation processes and their timescales, affect load curve shape, rate dependency and dissipation. This research will undoubtedly lead to LCE devices of even greater performances than those reported here. At the same time, there is evidently much richness yet to be explored in understanding and exploiting the anisotropy-controlled buckling deformations of LCEs. One can easily envisage how such deformations could be used to further optimize the nature of LCE deformations for a wide range of impact scenarios, for instance, in controlling rotational accelerations associated with oblique impacts.

In short, we have shown here that coupling the long-famed anisotropic non-linearity of LCEs with DIW 3D printing opens enormous application potential of impact absorbing LCE devices, with much physics and mechanics yet to be explored.

Here, we have demonstrated optimized DIW 3D printing of LCEs which we used to create the largest known monodomain devices to date. By testing these in compression over eight decades of strain rate—spanning quasi-static and impact rates—we have shown how the monodomain soft elastic mechanism enhances an elastomer's capability to load and dissipate mechanical impact energy. The monodomain soft elastic LCEs demonstrated the closest stress-strain response to that of an ideal absorber and its rate dependency led to a relatively consistent

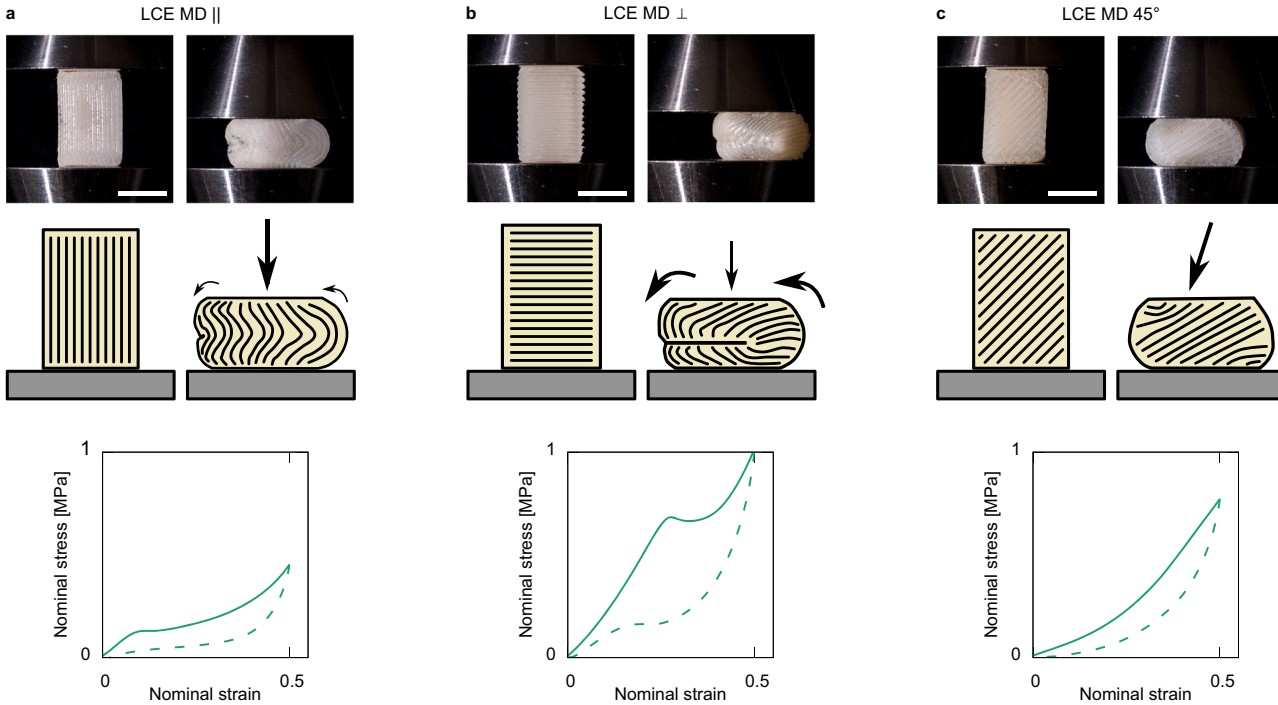

**Fig. 5 Anisotropy-controlled buckling.** Compression tests on high aspect ratio (height > in cross-sectional dimension) monodomain LCE devices with the director oriented (**a**) parallel, (**b**) perpendicular and (**c**) at 45° to the compression axis. Tuning the print axis in these devices enables further control of the buckling response and load curve shape of the DIW printed LCE devices. The figures show photographs to devices in their strained and unstrained states along will propose illustrations of the overall director profile in each state—traced from the print lines visible in each device, and the compressive nominal stress—nominal strain load curves recorded for each test. Bars are 10 mm.

performance over drop tests of differing intensities. We also showed how compressive deformations can be further enhanced by using the DIW printing direction to control the buckling characteristics of the LCE. This work brings the realization of LCE devices closer as it demonstrates a scalable and industrially realistic fabrication method along with enhanced material behavior not seen in conventional elastomers.

## Methods

**Materials and oligomer synthesis.** Structures for chemicals **3-7** below are shown in Supplementary Fig. 3. Acrylate-capped LC oligomers were synthesized using 4-(3-acryloyloxypropyloxy)benzoic acid 2-methyl-1,4-phenylene ester (RM257, CAS 174063-87-7), 2,2′-(ethylenedioxy)diethanethiol (EDDT, **3**, CAS 4970-87-7), butylated hydroxytoluene (BHT, **4**, CAS 128-37-0), 2-Hydroxy-4′-(2-hydroxyethoxy)-2-methylpropiophenone (HHMP, **5**, CAS 106797-53-9) and N,N,N′,N″,N″-Pentamethyldiethylenetriamine (PMDETA, **6**, CAS 3030-47-5). Polydomain LCEs were synthesized with the same chemicals, but with the addition of pentaerythritol tetra-kis(3-mercaptopropionate) (PETMP, **7**, CAS 7575-23-7). The non-mesogenic acrylate-capped oligomers were synthesized with the same components as the LC oligomer, but with RM257 replaced with the non-mesogenic diacrylate bisphenol A dimethacrylate (BPADMA, CAS 3253-39-2). RM257 was purchased from Wilshire Technologies, all other components were purchased from Sigma Aldrich and all components were used as received. Sorbothane Duro 70 was purchased from Isolate it! And cut to size for testing.

Liquid crystalline oligomers were synthesized via a base-catalyzed thiol-Michael click-reaction, described in detail elsewhere[15]. Briefly, BHT (radical-inhibitor, 0.300 g, 1.36 mmol), RM257 (diacrylate mesogenic monomer, 30.00 g, 49.7 mmol) and HHMP (UV-radical photoinitiator, 0.415 g, 1.85 mmol) were added to a glass vial and melted together in a water bath set at 70 °C. The melted components were thoroughly mixed, and bubbles/dissolved gases removed via vacuum before EDDT (dithiol spacer monomer, 8.30 g, 43.2 mmol) and PMDETA (base catalyst, 0.20 g, 1.17 mmol) were added and the mixture again mixed and degassed. The mixture was then transferred to the DIW-printing barrels and left in an oven set at 70 °C for half an hour to start the Michael addition (Supplementary Fig. 3). The barrel was then left at ambient temperature and protected from light for 2 days before printing. The chosen ratio of RM257:EDDT (1.000:0.870) ensured oligomers were acrylate capped and therefore would undergo crosslinking during 3D printing.

The LCE PD was synthesized via a similar method and composition, with the addition of the tetra-functional thiol crosslinker. We used a material of mol. ratio RM257 (5.00 g, 8.29 mmol):EDDT (1.38 g, 7.21 mmol):PETMP (0.28 g, 0.54 mmol) = 1.000:0.871:0.065 as this had an even balance of acrylate groups to thiol groups – ensuring that after the Michael addition (assuming complete conversion) no excess acrylate or thiol groups remained. In this synthesis, we used 0.052 g, 0.235 mmol of BHT and 0.072 g, 0.319 mmol of HHMP. PETMP was added at the same time as the EDDT and once all components were combined together, the mixture was poured into molds as opposed to the printing barrels.

For the synthesis of the non-liquid crystalline BPA-ink, the same oligomerization process used for the LC oligomer was used except for the following differences. First the non-mesogenic diacrylate monomer BPADMA was used in place of RM257 and in a ratio of BPADMA (20.00 g, 53.56 mmol):EDDT (9.88 g, 51.50 mmol) = 1:0.96. Second, we used 1.05 g, 4.78 mmol of BHT, 1.072 g, 7.48 mmol of HHMP and 0.849 g, 4.90 mmol of PMDETA.

Given the relatively low number of monomer units in each oligomer chain, small variations in measured material quantities translate to significant variations in average oligomer chain length. Therefore, to ensure comparable samples for each test, all materials of each type were prepared from single batches of prepared material.

**Direct-ink writing 3D printing.** DIW printing was performed using a Hyrel Engine HR 3D printer equipped with a KRA-2 print head for heating and extruding LC oligomers along directed print paths. Barrels containing printable oligomer were installed in the KRA print head which was set at 65 °C for the LC-ink (left for an hour prior to printing for equilibration) and kept at ambient conditions for the BPA-ink (due to its significantly lower viscosity). During printing, materials were extruded through a Tecdia Arque-S 5060 nozzle which had an internal diameter of 500 µm at the nozzle tip. G-code toolpaths controlling the print head's motion, printer settings and volumetric rate of material extrusion were created using in-house developed python scripts which also aided tuning of the print parameters (described below). During extrusion, the extruded material was exposed to UV light from LEDs surrounding the nozzle. Post-printing, the devices (typically 12 × 8 × 8 mm) were fully cured through exposure to high-intensity UV light in a UVP CL-1000 (Ultraviolet Crosslinkers, Upland, CA, USA) chamber for 2 h. Devices used in tensile stress-strain and DMA tests had far greater surface area-to-volume ratios and so were post-cured for 30 min. All printed devices were periodically rotated during post-curing to ensure even exposure.

Print conditions for our LCE were optimized through printing a series of matrices tuning, in turn, the various print parameters. First, we printed meanders of single lines - simultaneously optimizing for the volumetric extrusion rate and the nozzle height above the print surface. Liquid crystalline alignment quality was assessed via polarizing microscopy, with the parameters offering the greatest

apparent uniformity in, and contrast between the bright and dark states chosen as the ideal parameters. Next, using these parameters we printed series of meandering lines of different spacing between print lines until the print lines were close enough to bond to each other from a single printed sheet. Care was taken not to print lines too close to one another—which would diminish the level of liquid-crystalline alignment present. By measuring the thickness of the printed sheets, we deduced the ideal layer height to use for multi-layered devices.

**Gel fraction tests**. Tests were performed on 9 samples (~4 × 4 × 2 mm³ and ~0.06 g) of DIW-printed LCE, cut a from a larger 20 ×20 × 37 mm³ printed block (as illustrated in Supplementary Fig. 5). Using a mass balance of accuracy 0.1 mg, the mass of each sample ($m_i$) was recorded and then each sample was placed in a full 12 ml vial of toluene (a good solvent of LCEs) for 72 h. Samples were then removed and placed under vacuum in an oven set at 70 °C. Sample masses were measured periodically and were found to be constant after 48 hours of drying. The remaining mass of each sample after 48 h of drying ($m_f$) was measured again and the gel fraction, GF, calculated via, GF=100× $m_f/m_i$. From these tests the gel fraction of DIW printed LCEs was calculated to be 99.3 ± 0.4%.

**Dynamic mechanical thermal analysis**. Strips of monodomain LCE and BPA-ink elastomers were cut from larger printed sheets of printed layers, with the long edge either parallel or perpendicular to the print orientation as necessary. Strips had dimensions of ~25x5x1 mm with the gauge length being ~ 15 mm once clamped. Additionally, strips of polydomain LCE were cut from molded sheets of ~1 mm thickness.

Iso-frequency DMA temperature sweeps of the LCEs and BPA-ink elastomer was performed using a TA Instruments DMA Q800 equipped with an ACS-2 refrigerated air supply. Samples were loaded with a 0.01 N preload force (force tracking at 120% enabled) and were subject to 1 Hz oscillations of 0.1% strain amplitude. Samples were heated to 130 °C and allowed to equilibrate for 10 min to erase their thermal history before data were collected during a temperature sweep to −30 °C at 2 °C min⁻¹. Each test was repeated twice.

**Slow/intermediate rate mechanical and tan(δ) testing**. Tensile mechanical tests of the printed LCEs (upper right quadrant of Fig. 1h) were performed at room temperature using a TA Instruments DMA Q800 in displacement ramp mode. Strips of LCE, cut from larger printed sheets and of ~2.25 × 0.9 mm in cross-section and ~10 mm in gauge length once clamped, were stretched at a nominal strain rate of 1.6 × 10⁻⁴ s⁻¹.

Compressive mechanical tests (lower left quadrant of Fig. 1h, and Figs. 2c, 3) were performed using a TA Instruments Electroforce 3230 equipped with a ± 450 N load cell. Printed LCE and BPA-ink, LCE PD and SD-70 devices of dimensions ~12 × 8 mm² cross-sectional area and 8 mm height were prepared and in the case of printed devices, lightly sanded after freezing to give flat surfaces. The sample dimensions following sanding were used in analysis.

For comparisons of each material's compressive load curve and rate dependency, each device was tested by application of an initial 10 kPa preload stress (~1 N and loaded at 0.01 N s⁻¹) to ensure the platen was in contact with the top of the sample. After dwelling at this position for 600 s, the samples were loaded to 50% nominal strain and unloaded 0% to strain at strain rates of 10⁻⁴ (taken as quasi-static), 10⁻³, 10⁻², 0.1 and 1 s⁻¹. From the data collected, the dissipated strain energy density (area between loading and unloading curves) and percentage dissipated energy (area between load curves relative to area under loading curve) were calculated. For display in Fig. 3, the load curves of repeated tests (n ≥ 3) were averaged.

Measurements of the loss tangent, tan(δ), at different strains across each material's compressive load curve were performed using the Electroforce's DMA Application. Samples were loaded with a series of incremental forces (chosen based on each material's stiffness to 50% compressive strain). At each load increment, the samples were allowed to stress relax for 10 min before being subjected to 0.1% strain amplitude oscillations for DMA tests. Using each incremental load, the sample strain was extracted using the quasi-static compressive load curves measured for each material. Each material was repeat tested 5 times.

The buckling characteristics of higher aspect ratio (8 × 8 mm², cross-sectional area and 12 mm height) printed devices were assessed by subjecting the samples to a 10 kPa preload force (loaded at 0.01 N s⁻¹) and then loaded to 50% and unloaded at a rate of 10⁻³ s⁻¹. Photographs were taken of each sample in their unstrained and maximally strained states and from which the director orientation (on the face seen by the camera) could be traced using the print lines which were visible.

**Kolsky bar testing**. A Kolsky (also called split-Hopkinson) bar was used to measure the high nominal strain rate compression response of the materials beyond 1 s⁻¹. The Kolsky compression bar setup is composed of three axially aligned rods, the striker, incident, and transmission bars. The specimen is sandwiched between the incident and transmission bars in a stress-free state prior to the start of the experiment. The Kolsky bar is actuated when the striker bar is accelerated using compressed gas from a gun barrel. When the striker bar impacts the end of the incident bar, an incident stress wave is generated that propagates along

the bar until it reaches the sample. The specimen is compressed at a high deformation rate when the incident wave arrives at the end of the incident bar. Part of the wave is transmitted through the sample into the transmission bar, and part is reflected in the incident bar as a reflected wave. Strain gages mounted on the incident and transmission bars allow measurement of the specimen nominal stress, strain, and strain rate according to ref. [29]. Pulse shaping is a critical step in achieving a constant nominal strain rate deformation and an equilibrated stress in the sample, both of which are required for a valid experiment. Small disks of annealed copper, or "pulse shapers" are placed on the impact end of the incident bar. The dimensions of the disks are designed to vary the profile of the incident pulse to achieve constant strain rate in the sample.

The dimensions of the Kolsky compression samples were approximately 2 × 5 × 5 mm³ (2 mm thick), dimensions which aided the stress equilibration process. Experiments were carried out at nominal strain rates of 800, 1600, and 3000 s⁻¹ with n = 3 at each condition. Samples were subjected to a minimum of 50% nominal strain for comparison with quasi-static experiments. For display in Fig. 3, the load curves of repeated tests were averaged.

The supplementary information provides further details showing the specimens reached constant strain rate, dynamic equilibrium, and were not affected by radial inertia or friction. The supplementary methods cite refs. [30–33].

**True stress relative to ideal absorber**. First, for each material and at each nominal strain rate (using their averaged load curves), the amount of loaded strain energy, $E_L$, was calculated by numerically integrating the averaged load curves, $\sigma(\epsilon)$, between true strains of 0 and 0.7,

$$E_L = \int_0^{0.7} \sigma(\epsilon)\mathrm{d}\epsilon \qquad (3)$$

An ideal absorber would instead absorb this energy over this strain range at a constant stress level of $E_L/0.7$. Therefore, the original load curve can be normalized against the stress level of an ideal absorber by plotting $(\sigma(\epsilon)\times 0.7)/E_L$, as is shown in Fig. 3c.

**Drop testing**. Drop tests were performed using an in-house built test frame. A uniaxial piezoresistive accelerometer (Meggit model 7264B-2000) was used for measuring linear accelerations. Data were acquired at a sample rate of 25k Hz using a National Instruments 9237 data acquisition bridge module. The accelerometer was attached to a cylindrical 2 kg mass (38 mm radius) mounted on a guided linear rail that was dropped on samples of typical dimensions 12 × 12 mm² cross-sectional area and 7.5 mm height. Each material was subjected (at n = 3 for each condition) to drop tests with the mass dropped from heights of 0.25, 0.50, 0.75, and 1.00 m (due to its softness, SD-70 was not subjected to the drop from 1.00 m). Table S1 shows, for each drop height, the impact speed, initial nominal strain rate and impact energy density. Accelerometer data was passed through a channel frequency class 1000 filter according to SAE J211 and data was cropped to capture once a change in acceleration of +1 g (=+9.812 m s⁻²) was detected—a displacement position taken as the top of the impacted sample. The impact velocity was deduced from a separately determined calibration curve linking drop height to impact velocity. Continued displacement was calculated using the impact velocity and twice integration of the accelerometer's readings.

GSI for ideal absorbers, capable of constant deceleration to 100% compressive nominal strain, were calculated using the impact velocity and height of each sample to determine the ideal-impact time and deceleration value. The GSI of each sample at each drop height was compared to this value of the idealized absorber.

## Data availability

Raw data gathered for this work is available at the following https://doi.org/10.6084/m9.figshare.16592678. Any additional data can be sought from the authors.

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

## Acknowledgements

English Speaking Union through a Lindemann Fellowship (D.M.) and the Leverhulme Trust through an Early Career Fellowship grant number ECF-2020-68 (D.M.). This material was based upon work supported by, or in part by, the U.S. Army Research Laboratory and the U.S. Army Research Office under grant number W911NF1710165 (N.A.T). This work was also supported in part under NSF CAREER Awards CMMI-2046611 (K.Y.) and CMMI-1350436 (C.M.Y.), by the National Football League under the HeadHealthTECH program, and through the Laboratory Directed Research and Development program at Sandia National Laboratories. Sandia National Laboratories is a multimission laboratory managed and operated by National Technology & Engineering Solutions of Sandia, LLC, a wholly owned subsidiary of Honeywell International Inc., for the U.S. Department of Energy's National Nuclear Security Administration under contract DE-NA0003525. This paper describes objective technical results and analysis. Any subjective views or opinions that might be expressed in the paper do not necessarily represent the views of the U.S. Department of Energy or the United States Government. SAND2021-13516 J (B.S., B.S., K.N.L.)

## Author contributions

Research was conceived and directed by C.M.Y, D.M., K.Y and K.L. Experimental work and analysis was performed by D.M., N.A.T., B.Sanborn, R.H., L.C., R.Z. and B.Song. Manuscript preparation was led by D.M., C.M.Y. and N.A.T. All authors provided input on manuscript drafts

## Competing interests

C.M.Y., D.M. and L.C. have a potential conflict of interest since they own equity in companies that are trying to commercialize LCE products. The remaining authors declare no competing interests.
