## [Peer Review File · Nature Communications]

REVIEWER COMMENTS

Reviewer #1 (Remarks to the Author):

The paper presents some very interesting results about the energy dissipation capacity of liquid crystal elastomers. Bulk sized monodomain LCE samples are obtained through 3D printing method and are tested compressively for strain rates from quasi-static to 3000/s by using various testing methods. All results have shown that the LCE samples with soft elasticity exhibit strong energy dissipations. It has indicated clearly that the soft elasticity is also the main factor affecting the mechanical behavior of LCE under compressive high strain rate loadings. And it demonstrates possibilities of some real applications.

Some suggestion and comments for the authors.

1. Fig. 1 (h). The tensile and compressive mechanical anisotropy of printed LCEs. It is interesting to plot them together in one figure. However, I suppose that they (tensile and compressive tests) were done by using different samples and test machines, because I have not found any testing method to perform the tensile and compressive tests simultaneously in the paper.
2. Fig. 3c). It is difficult to understand what is “True stress relative to ideal absorber”. Maybe you can give a definition by using a math formula?
3. Fig. 4. It would be better if you can show the results for LCE MD \perp and PD samples as well.
4. Fig. 5. “director profile in each state – traced from the print lines visible in each device “. I think that this is not always the case. It should be correct at the beginning before loading because of the 3D printing process. After compression, the director fields inside the samples can rotate very strongly and very heterogeneously. Although there is no direct evident of director rotations for samples under compression due to the experimental difficulties, we do know for LCE samples under extension that stripe domains (thin layered regions with alternative rotated director fields) can form after some critical loads. And such domains can be very important for energy dissipations.
5. Line 209: “this implies viscous and plastic behavior”. What do you mean by “plastic behavior”? I understand that the dissipation due to the viscosity is obvious but is not enough to account for the observed behavior of LCE MD \parallel . However, it cannot be plastic, right? One possible mechanism could be the stripe domains as discussed in the following paper: Programmable Mechanical Energy Absorption and Dissipation of Liquid Crystal Elastomers: Modeling and Simulations, First published: 29 July 2021 in Advanced Engineering Materials (<https://doi.org/10.1002/adem.202100590>)
6. Method: Kolsky bar testing. Some more details are appropriate. You may add them in your supporting information file. 1) It is very important that the SHPB test has reached the stress equilibrium condition and constant strain rate condition. So, please provide the material and size information of bars and pulse shapers. Please provide the strain rate curves of each stress-strain curve to demonstrate that the tests reached constant strain rate condition, and the stress wave has oscillated at least three times in the specimens before the incident wave arrival plateau stage. 2) Since the LCEs could bear large deformation during the SHPB tests, the samples are designed as 5X5X2 mm³ in dimension, which is very thin, might generate large transverse deformation and friction when testing by SHPB. Then the SHPB results might be incorrect. So please provide some evident that the transverse inertia effect and the friction effect can be neglected.

7. Some print errors: 1) Line 84: “bi-stip devices” should be bi-strip? 2) In the caption of Fig. 3c): “Representative curves are shown from a testing size of. (d)”? Some words are missing? 3) Line 286: “sharp peaks in, and high values of acceleration”. Some words are missing? 4) Line 506: “Diving this by a true strain of” should be “Dividing”?

Reviewer #2 (Remarks to the Author):

Mistry et al. describe the dissipation of compressive loads by 3D printed, oriented LCEs. The anisotropic nature of the LCEs leads to seeming improvements in dissipation of energy in compression as compared to a chemically similar isotropic elastomer and a commercially available material. The work will be of interest to researchers in the field of LCEs and those in the fields of dissipative materials. However, there are some significant concerns that should be addressed.

One major concern is that the data are not presented with any indication of the level of error. In the methods section, the authors state that replicates were performed. In many figures, ‘error bars’ should be added. It is impossible to verify the conclusions of the paper without some accounting for experimental and sample variability.

Another major concern is that there is significant porosity in the LCE samples. How can we reasonably assume that this porosity (18%) is not important, especially at high compressive strains? I note that the authors did consider the compressive behavior of an isotropic elastomer, but that material has only 9% porosity. The characterization of that isotropic elastomer also raises other questions, the authors state that “We note that the porous channels do not introduce any anisotropy in the character of the BPA material’s dynamic and quasi-static mechanical behavior”. This seems to contradict the results in Figure S2, where the quasistatic compressive behavior is quite anisotropic. It seems likely that an additional control is needed where the porosity of the LCE sample is matched in the BPA structures. The anisotropy of the BPA samples should also be considered.

A major problem with photopolymerization of thick objects is that postcuring the center of such a structure is quite difficult. Have the authors verified (or indirectly supported) that the center of the objects is indeed similar in crosslink density to the outer regions using DMA, DSC, gel fraction, or other measurements? Notably the isotropic elastomer is likely much more transparent than the LCE. This difference could magnify the differences between the materials.

As a minor comment, the authors state “Second, the apparent magnitude of anisotropy is less than that seen in tension, an intuitive result given the maximum nominal strain of -1 in compression.” It is not clear to me how this is an intuitive result.

Dear Reviewer,

Thank you for your positive and constructive criticisms of our manuscript. We have thought carefully about the points you have raised and have addressed them in our revised manuscript and in the point by point response below. We hope you find that our modifications address the points you have raised, and we thank you for the time you have spent reviewing our manuscript.

Best wishes,

Dr Devesh Mistry

Reviewer #1

The paper presents some very interesting results about the energy dissipation capacity of liquid crystal elastomers. Bulk sized monodomain LCE samples are obtained through 3D printing method and are tested compressively for strain rates from quasi-static to 3000/s by using various testing methods. All results have shown that the LCE samples with soft elasticity exhibit strong energy dissipations. It has indicated clearly that the soft elasticity is also the main factor affecting the mechanical behavior of LCE under compressive high strain rate loadings. And it demonstrates possibilities of some real applications.

We thank the reviewer for their positive comments on our manuscript and are glad to read that they have found our manuscript interesting.

Some suggestion and comments for the authors.

1. Fig. 1 (h). The tensile and compressive mechanical anisotropy of printed LCEs. It is interesting to plot them together in one figure. However, I suppose that they (tensile and compressive tests) were done by using different samples and test machines, because I have not found any testing method to perform the tensile and compressive tests simultaneously in the paper.

The reviewer is correct that the tensile and compressive tests were separate tests and are plotted together for easy visual comparison of the respective behaviours. To make it clearer that these tests were separate we have added a note to our figure 1 caption and in the main text at line 90/91.

2. Fig. 3c). It is difficult to understand what is “True stress relative to ideal absorber”. Maybe you can give a definition by using a math formula?

As suggested we have provided additional details of how the true stress relative to the ideal absorber was calculated in our methods section - True stress relative to ideal absorber (lines 525-532).

3. Fig. 4. It would be better if you can show the results for LCE MD \perp and PD samples as well.

We have now provided **supplementary figure 4** which includes data for the LCE MD \perp and PD as well. We had originally included them in the main text figure, but found that the number of graphs became overwhelming and it became more difficult to appreciate the differences that we are

highlighting in this manuscript. From figure 3 and its discussion, the advantages of (the soft elastic) LCE MD \parallel over (the non/minimally soft elastic) LCE MD \perp and PD are made clear. Therefore, in figure 4 we felt it is made most sense to just focus on LCE MD \parallel and the isotropic materials.

4. Fig. 5. “director profile in each state – traced from the print lines visible in each device “. I think that this is not always the case. It should be correct at the beginning before loading because of the 3D printing process. After compression, the director fields inside the samples can rotate very strongly and very heterogeneously. Although there is no direct evidence of director rotations for samples under compression due to the experimental difficulties, we do know for LCE samples under extension that stripe domains (thin layered regions with alternative rotated director fields) can form after some critical loads. And such domains can be very important for energy dissipations.

What the reviewer has discussed is indeed a rich area to study in itself! We agree that there is a possibility that stripe domains may have formed in the sample compressed parallel to the director – although we are not aware of any stripe domains (like those reported by Finkelmann & co.) being recorded and published for thiol-acrylate or other click-chemistry LCEs. As the reviewer highlights, quantifying and measuring this effect would be incredibly hard – if not impossible – for the sample dimensions needed for compressive tests.

We have amended the caption to say that the illustrations are “proposed” for the “overall” director orientation. We have also added a short discussion near line 335 noting that the lines shown are what we would propose as the orientation averaged over any stripe domains that may be present. We hope these clarifications are agreeable and mean that it is clearer that the figure is demonstrating how different the deformation and buckling behaviour can be when controlling the direction of anisotropy in the printed devices.

5. Line 209: “this implies viscous and plastic behavior”. What do you mean by “plastic behavior”? I understand that the dissipation due to the viscosity is obvious but is not enough to account for the observed behavior of LCE MD \parallel . However, it cannot be plastic, right? One possible mechanism could be the stripe domains as discussed in the following paper: Programmable Mechanical Energy Absorption and Dissipation of Liquid Crystal Elastomers: Modeling and Simulations, First published: 29 July 2021 in Advanced Engineering Materials (<https://doi.org/10.1002/adem.202100590>)

This is quite an interesting point! In this sentence we were attempting to describe how the behaviour has some plastic characteristics - in that on a short time scale (1s – 10s seconds), applied strains are persistent due to a slow relaxation and the strain energy being largely dissipated. Evidently, some strain energy does remain because the samples elastically recover to their initial state.

To remove any confusion about any plasticity, or plastic nature that may be present we have modified the statement to read “*this implies viscous and somewhat liquid behaviour*”. We hope this clarification conveys the message we had originally intended.

6. Method: Kolsky bar testing. Some more details are appropriate. You may add them in your supporting information file. 1) It is very important that the SHPB test has reached the stress equilibrium condition and constant strain rate condition. So, please provide the material and size information of bars and pulse shapers. Please provide the strain rate curves of each stress-strain curve to demonstrate that the tests reached constant strain rate condition, and the stress wave has oscillated at least three times in the specimens before the incident wave arrival plateau stage.

2) Since the LCEs could bear large deformation during the SHPB tests, the samples are designed as 5X5X2 mm³ in dimension, which is very thin, might generate large transverse deformation and friction when testing by SHPB. Then the SHPB results might be incorrect. So please provide some evident that the transverse inertia effect and the friction effect can be neglected.

Thank you for the comments on the high rate experiments. We have added a lengthy explanation addressing your concerns in the revision in the supporting information file. We showed that the specimens reached constant strain rate, dynamic equilibrium, and were not affected by radial inertia or friction.

7. Some print errors: 1) Line 84: “bi-stip devices” should be bi-strip? 2) In the caption of Fig. 3c: “Representative curves are shown from a testing size of. (d)”? Some words are missing? 3) Line 286: “sharp peaks in, and high values of acceleration”. Some words are missing? 4) Line 506: “Diving this by a true strain of” should be “Dividing”?

Many thanks to the reviewer for highlighting these errors – we have corrected them.

Reviewer #2 (Remarks to the Author):

Mistry et al. describe the dissipation of compressive loads by 3D printed, oriented LCEs. The anisotropic nature of the LCEs leads to seeming improvements in dissipation of energy in compression as compared to a chemically similar isotropic elastomer and a commercially available material. The work will be of interest to researchers in the field of LCEs and those in the fields of dissipative materials. However, there are some significant concerns that should be addressed.

One major concern is that the data are not presented with any indication of the level of error. In the methods section, the authors state that replicates were performed. In many figures, ‘error bars’ should be added. It is impossible to verify the conclusions of the paper without some accounting for experimental and sample variability.

We appreciate the reviewer’s highlighting the missing errors – of the relevant figures, errors are shown for figure 3 d and e (they are small and only clearly visible for one point in figure 3e for the BPA ink at a nominal strain rate of 10^0 s^{-1}) but they were not included in figures 3f and 4c. For figure 4c their inclusion made it difficult to distinguish the different point types indicating the different drop heights. We have added the error bars again and we believe we have found a way to ensure the different point types are clear. For figure 3f, the error bars are smaller than the data points are shown:

If readers wished to analyse our raw data, then it is provided in the following dataset: <https://figshare.com/s/ec7d4e9f0fee688a5a2b>. Upon publication of the manuscript this dataset will be made public and minted with the DOI 10.6084/m9.figshare.16592678.

For the in-house prepared, thiol-acrylate materials (printed LCE MD, and BPA-ink, and moulded LCE PD), the sample variability is difficult to account for and report without overshadowing the basic mechanical differences that we are highlighting in this work.

For instance, the printed LCE is ideally formed from RM257 and EDDT in a molecular ratio of 1.000:0.870. According to Carothers's equation, this gives an average oligomer chain length of 14.33 (assuming 100% conversion). On the scale that we (and others in the community) can currently produce these materials for DIW printing, chemical measuring errors of ~2% are unavoidable. However, this acceptable level of measurement error would translate to an average oligomer chain length of 12.76 which is 12% less than the target. Upon network formation, this would result in a material with a much different crosslink density, incomparable with materials of perfect chemical composition. To ensure that all of our samples prepared for each test were the same and comparable against each other in the data presented, we produced all samples of each material from single synthesis batches.

The data presented in our manuscript and the repeat tests performed are therefore the clearest accounts we can provide to demonstrate the mechanical differences offered by the soft elastic effect. Our conclusions are verified both by the low experimental errors and by the agreement of our data with others from the LCE field – (i.e. base tests from figure 1 are consistent with the literature).

In our manuscript methods, we have added a paragraph outlining how samples were produced from single batches of material and why this was necessary. We hope that this adds sufficient context to our results for the reader to follow our conclusions.

Another major concern is that there is significant porosity in the LCE samples. How can we reasonably assume that this porosity (18%) is not important, especially at high compressive strains? I note that the authors did consider the compressive behavior of an isotropic elastomer, but that material has only 9% porosity. The characterization of that isotropic elastomer also raises other questions, the authors state that “We note that the porous channels do not introduce any

anisotropy in the character of the BPA material's dynamic and quasi-static mechanical behavior". This seems to contradict the results in Figure S2, where the quasistatic compressive behavior is quite anisotropic. It seems likely that an additional control is needed where the porosity of the LCE sample is matched in the BPA structures. The anisotropy of the BPA samples should also be considered.

This comment highlights several important aspects which we, in our experimental design, had to balance. What we will state first, is that this manuscript isn't purely about the mechanics and physics of LCEs, but it's about the mechanics and physics of 3D printed LCEs. As we stated in our introduction. DIW printing is currently the only way we (researchers/engineers) have to controllably produce bulk monodomain LCE structures of high molecular order and in non-trivial geometries.

Key to the molecular order and size of the samples produced is the viscosity of the LCE ink. Sufficiently high viscosity was required for high molecular order in devices of many printed layers. The consequence of this, was the formation of the air channels which resulted in the 20% porosity. In our methods tuning, we found that reducing this porosity by changing for instance viscosity or printing parameters, leads to a reduction in order and essentially polydomain devices. Details of all the printing parameters we have considered in our printing are highlighted in the following paper: <https://aip.scitation.org/doi/full/10.1063/5.0044533>

This is one reason why we needed to prepare a comparator isotropic and DIW printable material.

However, designing an equivalent material of similar chemical structures (main chain polymer containing benzene rings) and similar thermomechanical properties (most notably similar glass transition temperatures and storage moduli) was only possible in a material with a significantly lower ink viscosity (a basic consequence of the ink being an isotropic polymer and not a nematic polymer). With this lower viscosity ink, we could not produce a printed material with an identical porosity – 9% being the maximum possible using our 3D printer.

Thus, we believe that on balance, the material we produced as a comparator is as similar as reasonably possible given the number of characteristics we had to try and match. If we were to match the porosity, then the thermomechanical properties of the comparator material would be far different than our LCE's.

However, our results show:

- 1) that the behaviour of the monodomain LCE is consistent with the solid monodomains tested under compression produced by Shaha et al. (manuscript Ref. 12, noted in line 44) – therefore the anisotropic behaviour we report can be reasonably attributed to the soft elasticity of LCEs.
- 2) The DMA tests on the isotropic BPA-ink printed devices show no anisotropy – so any anisotropy in dynamic tests of our LCE is unlikely to be a consequence of the porous anisotropy.

While, the printed BPA material does show anisotropy in the modulus, it shows isotropy in the load curve shape – they are both still the same overall shape. Given the porous anisotropy, it is not surprising to see a difference in modulus for the isotropic BPA-ink elastomer. But, in terms of this manuscript, what is most important is that the character of the loading and unloading curves show isotropy. By contrast, the differences in character for the monodomain LCE is clearly very much different and fits very well with the expected differences from the literature.

In short, the characterisations provided here are on 3D printed materials – i.e. the material and its additive processing. We have taken numerous steps to provide adequate comparator materials (a comparator polydomain LCE, an isotropic DIW printed elastomer and a commercial dissipative material) and draw sensible conclusions from our data that we are confident in presenting. We strongly believe that adding additional data for both orientations of the BPA sample would detract from the key results of this paper, and is unnecessary given they are both the same in character.

A major problem with photopolymerization of thick objects is that postcuring the center of such a structure is quite difficult. Have the authors verified (or indirectly supported) that the center of the objects is indeed similar in crosslink density to the outer regions using DMA, DSC, gel fraction, or other measurements? Notably the isotropic elastomer is likely much more transparent than the LCE. This difference could magnify the differences between the materials.

We have performed gel fraction tests on our 3D printed LCEs and found that for nine $4 \times 4 \times 2 \text{ mm}^3$ samples taken from the various locations near the centre of a $20 \times 20 \times 37 \text{ mm}^3$ printed device, the gel fraction was $99.3 \pm 0.4\%$. These data confirm proper curing of our printed devices. These details have been added to the manuscript in lines 73/74, lines 446-454 (methods) and **supplementary figure 5**.

As a minor comment, the authors state “Second, the apparent magnitude of anisotropy is less than that seen in tension, an intuitive result given the maximum nominal strain of -1 in compression.” It is not clear to me how this is an intuitive result.

Here we are referring to the fact that in compression, you cannot compress a solid material to 100% (a liquid would just flow away). So irrespective of the softness of the material at low strains, the stresses will always tend to infinite for a nominal strain of -1. By comparison, in tension a soft material can be hyperelastic and can stretch to many hundreds of percent at a low stress level and stiffer materials may only stress to a strain of, say, 0.5 before failure. As such, anisotropy would be expected to be much less evident in compression than in tension. In figure 1h this exactly what we see. In light of the reviewer’s comment we have clarified the statement in the manuscript (line 97)

REVIEWERS' COMMENTS

Reviewer #1 (Remarks to the Author):

The authors have addressed all previous questions.

No further comments.

Reviewer #2 (Remarks to the Author):

The authors have revised the manuscript to address reviewer concerns. I continue to believe that this work will be of interest to researchers in several areas. As such, I recommend publication in the current form.